# Circadian Variations and Associated Factors in Patients with Ischaemic Heart Disease

**DOI:** 10.3390/ijerph192315628

**Published:** 2022-11-24

**Authors:** Marisa Estarlich, Carmen Tolsa, Isabel Trapero, Cristina Buigues

**Affiliations:** 1Department of Nursing, University of Valencia, 46010 Valencia, Spain; 2Epidemiology and Environmental Health Joint Research Unit, FISABIO-Universitat Jaume I, Avenida de Catalunya 21, 46020 Valencia, Spain; 3Spanish Consortium for Research on Epidemiology and Public Health (CIBERESP), 28001 Madrid, Spain; 4Frailty and Cognitive Impairment Group (FROG), University of Valencia, 46010 Valencia, Spain

**Keywords:** myocardial infarction, circadian rhythm, individual risk factors, AMI location, severity

## Abstract

Circadian rhythms have been identified in cardiovascular diseases, and cardiovascular risk factors can modify the circadian rhythm. The purpose of this study was to describe the onset of ischaemic heart disease symptomatology in relation to the date and time, the day of the week of presentation, the season, AMI location and severity and the level of influence of individual patient characteristics in a retrospective cross-sectional study involving 244 ischaemic heart disease patients from the intensive care unit of La Ribera Hospital (Spain). The onset of pain was more frequent in the morning, the season with the highest frequency of ischaemic events was winter, and the lowest incidence was during weekends. Regarding the severity of ischaemic heart disease, the circadian rhythm variables of weekdays vs. weekends and seasons did not show a significant association. The length of hospital stay was associated with the onset of pain in the afternoon. The onset of pain at night was associated with the subendocardial location of the infarction. In conclusion, living in a Mediterranean country, the Spanish population showed a circadian pattern of AMI, where the onset of pain has an influence on AMI location and on the length of hospital stay and is the same in patients with different individual risk factors.

## 1. Introduction

Circadian rhythms related to day and night have been identified in multiple physiological processes that may affect cardiovascular diseases. This rhythm is synchronised with the variable (circadian) production of hormones, neurotransmitters and cytokines, as well as the activation of various brain areas and the main functions of key organs, such as the pancreas, liver, adrenal glands, intestine, lungs and heart [1,2].

Several studies evaluating chronobiology have established that there is circadian and seasonal periodicity in the incidence of ischaemic heart disease [3]. Almost all the cardiovascular variables evaluated at different times of the day and night have day–night patterns, including blood pressure, heart rate, circulating catecholamines, markers of blood clotting and vascular endothelial function [4,5]. Cardiovascular disease has been found to follow a circadian rhythm throughout the entire day rather than following random manifestations [6,7,8].

Cardiovascular diseases are the most frequent cause of death in modern society [9,10]. Mortality is related in a large number of cases to the progression of deteriorating cardiac function or the triggering of malignant arrhythmias, such as ventricular tachycardias or ventricular fibrillation [11,12].

Historically, continuous efforts have been made to gain a better understanding of the main cardiovascular risk factors and acute coronary syndrome (ACS) in an attempt to expand existing knowledge on the clinical, pathophysiological, epidemiological and therapeutic aspects of the disease [13,14,15]. Recent advances in chronobiological methodology have provided new insights into the behaviour of rhythmic phenomena in different cardiovascular risk factors that had not been considered in earlier studies [3,6,7,16].

In recent years, previous studies have suggested the existence of variability in the timing of cardiovascular events, with an irregular distribution of myocardial ischaemic episodes [17,18,19,20]. All showed a non-uniform distribution, with increases in incidence during certain hours, days of the week or months.

Ischaemic heart disease episodes can occur at any time of the day; however, circadian variations in their onset have been found in the general population [21,22,23]. Some cardiovascular risk factors, such as diabetes and hypertension, are associated with chronotype and sleep, whereas diabetes, smoking and reinfarction can modify the circadian rhythm. The circadian desynchronisation of the body in its environment, either through rotating shift work schedules or genetic alterations, increases the development of cardiovascular diseases.

As one of the most important factors in cardiovascular disease management is recognising individual risk factors [24], we question whether these individual risk factors can be influenced by circadian variables. Therefore, this research proposal aims to help elucidate the mechanisms underlying the periodicity of myocardial infarction presentation by describing the onset of ischaemic heart disease symptomatology in relation to the time of presentation, annual season and whether it is influenced by patient characteristics that may be determinants of morbidity.

## 2. Materials and Methods

### 2.1. Design and Setting

This was a retrospective cross-sectional study that included patients diagnosed with ischaemic heart disease who were referred to the intensive care unit (ICU) of La Ribera Hospital.

The La Ribera Hospital is in the town of Alzira (Spain) and provides health care to a population of 257,593 inhabitants, according to the “Management Memorandum of the Regional Ministry of Universal Health and Public Health” of 2015.

The study was conducted in accordance with the principles set out in the Declaration of Helsinki, and approval was obtained from the Alzira Hospital Research Ethics Committee.

### 2.2. Study Variables

Patients belonging to the Alzira Health Department admitted to the ICU with a diagnosis of ischaemic heart disease (410–414/410—acute myocardial infarction; 411—other acute and sub-acute forms of ischaemic heart disease; 412—myocardial infarction; 413—angina pectoris; and 414—other forms of chronic ischaemic heart disease) according to the International Classification of Diseases (ICD-9) were included. The criteria of myocardial infarction in patients with a clinical presentation of acute coronary syndromes were based on those established by the European Guidelines for the management in patients with or without persistent ST-segment elevation [25,26] and stable coronary artery disease [27] based on the integration of low likelihood and/or high-likelihood characteristics derived from clinical presentation (i.e., symptoms, vital signs), 12-lead ECG, and cardiac troponin [25].

Following these guidelines, criteria for acute myocardial infarction were: symptoms of ischaemia, new or presumed new significant ST-T wave changes or left bundle branch block on 12-lead ECG.), development of pathological Q waves on ECG, imaging evidence of new or presumed new loss of viable myocardium or regional wall motion abnormality and intracoronary thrombus detected on angiography or autopsy [26]. Cardiac troponin levels were used for the diagnosis of unstable angina, and non-invasive imaging and laboratory biochemical tests for other ischaemic heart diseases [25].

#### 2.2.1. Circadian Rhythm Variables

The date and time of symptom onset were recorded to obtain the time, day and season of the year. The time variable was recoded into three categories: morning (6–12 h), afternoon (12–21 h) and evening (21–6 h). The season of the year was recorded according to the date of admission and discharge from the hospital: “Spring (19-03-year/21-06-year)”, “Summer (21-06-year/22-09-year)”, “Autumn (22-09-year/21-12-year)” and “Winter (21-12-12-year)”. The variable day of the week was codified as a workday (Monday, Tuesday, Wednesday, Thursday or Friday) or as a weekend day (Saturday and Sunday).

#### 2.2.2. Sociodemographic and Risk Factors

These are the variables of a sociodemographic nature and possible comorbidities. The sociodemographic variables considered in the analyses were age and gender. In addition, we used the possible comorbidities known at the time of admission, which were related to the following risk factors: smoking, hypertension, diabetes mellitus, dyslipidaemia, history of ischaemic heart disease and family history of ischaemic heart disease.

#### 2.2.3. Ischaemic Heart Disease Variables

Severity

The severity of heart failure (HF) was analysed according to the Killip classification [28]:

Killip class I: patients without any clinical sign of HF;

Killip class II: crackles or rales in the lungs, elevated jugular venous pressure and S3 gallop;

Killip class III: patients with evident acute pulmonary oedema;

Killip class IV: cardiogenic shock or hypotension (systolic blood pressure < 90 mmHg) and features of low cardiac output (oliguria, cyanosis or impaired mental status).

The last two severity categories were combined, as only six individuals were included in the Killip IV class.

Location of acute myocardial infarction

Patients designated with code 410. Xx (acute myocardial infarction and localisation) were recorded in three categories: anterior, inferior and subendocardial.

Length of hospital stay

The total number of days between ICU admission and in-patient hospital stay.

These variables were obtained by consulting the computerised clinical history of the operation at the Hospital de La Ribera through the SIAS program. Prior to this, appropriate access permissions were requested in accordance with the Data Protection Act at the time of the study.

### 2.3. Statistical Analysis

A descriptive study was conducted on the circadian rhythm, control and ischaemic heart disease variables. Continuous variables are represented by descriptive statistics (mean and SD values), and categorical variables, by counts (frequency/percentage).

The associations between circadian rhythm variables and the severity of ischaemic heart disease were assessed by means of logistic ordinal regression, the location of acute myocardial infarction multinomial regression and the length of hospital stay linear regression. Factors included in multivariate models were selected according to the following: first, bivariate model factors related to ischaemic heart disease outcome variables were considered (*p* < 0.1 in the likelihood ratio test); second, multivariate models were built backward in a stepwise manner, and the covariates were retained in the final model if they were related to the outcome (based on likelihood ratio (LR) tests with a *p*-value of <0.05). Age, sex and smoking were included in all models despite their statistical significance, following previous studies. The severity and location of acute myocardial infarction associations were expressed as the odds ratios (OR), confidence intervals at 95% (CI 95%) and the length of hospital stay as % change.

To detect a modification effect related to the covariate variables, we expanded our main models with an interaction term between the control variables that were significant in the main models and circadian rhythm variables. Thereafter, we compared our main models with and without the interaction term by applying the Wald test. In the case of finding a significant interaction, we stratified our main models according to the significant variables. Statistical analyses were conducted using R software, version R-4.0.5 (Foundation for Statistical Computing, Vienna, Austria).

## 3. Results

### 3.1. Study Population Characteristics

The study population consisted of 244 patients diagnosed with ischaemic heart disease at a known time of symptom onset. The participants were mostly males (70.9%), and the mean age (+SD) was 65.13 ± 13.41 years. A medical history of angina or AMI was present in 40.6% of patients, hypertension in 50%, diabetes mellitus in 27.9% and dyslipidaemia in 45.1%, and 46.7% of patients were non-smokers.

### 3.2. Ischaemic Heart Disease Variables

In terms of the type of ischaemic heart disease, more than half of the patients (66,4%) had an episode of acute myocardial infarction with the predominant inferior location of the AMI (40,1%). Patients classified as having Killip II were the most common (Table 1).

The factors associated with the severity of ischaemic heart disease in the model were age (OR = 1.03, 95% confidence interval (CI) 1.01–1.05), previous ischaemic heart disease (OR = 3.12, 95% confidence interval (CI) 1.83–5.32) and dyslipidaemia (OR = 1.69, 95% confidence interval (CI) 1.03–2.78) (Table 2).

### 3.3. Circadian Rhythm Variables

The onset of pain was more frequent in the morning (38.1%). The season with the highest frequency of ischaemic events was winter, and Monday, Tuesday, Wednesday and Thursday were the days of the week with the highest incidence compared with Friday to Sunday.

Regarding the factors associated with ischaemic heart disease variables related to circadian rhythm, the onset of pain in the afternoon or evening was associated with a lower risk of severity compared with participants in whom the pain started in the morning, although this was not significant. The circadian rhythm variables, weekdays vs. weekends and season, showed no significant association with the severity of ischaemic heart disease, although the risk of high-severity heart disease was greater for participants whose pain started at the weekend and was lower in all seasons other than in winter.

The length of hospital stay was associated with previous ischaemic heart disease (% change =1 6.55, 95% confidence interval (95%CI) 4.05–30.55), being a former smoker (% change = 19.38, [1.36, 40.59]) and with the onset of pain in the afternoon (% change = 26.33, [10.44, 44.52]). The weekdays vs. weekends variable and season did not show a significant association, although the % change on weekends was lower than on workdays and the % change in spring was less than in winter (Figure 1).

After selecting patients with localised AMI (*n* = 149), Table 3 presents factors associated with the location of the AMI. Age (OR = 1.3, 95% confidence interval (CI) 1.00–1.06) and being a smoker (OR = 2.82, 95% confidence interval (CI) 1.09–7.27) increased the likelihood of having an inferior AMI in relation to the anterior location. The onset of pain at night (OR = 6.87, 95% confidence interval (CI) 1.67–28.15) was associated with a subendocardial location of the infarct rather than an anterior location.

After introducing the interaction term in our analyses, we did not find a significant interaction between the circadian rhythm and covariate variables in any of the models. The circadian rhythm followed the same pattern in all patients, regardless of whether they were dyslipidaemic, diabetic, hypertensive or had another risk factor.

## 4. Discussion

In this paper, we investigated whether a circadian pattern in the occurrence of ACS exists and what factors influence the severity of AMI, its location, the length of hospital stay in patients in a Mediterranean country and whether individual risk factors result in differing patterns.

Environmental influences on the development and form of CVD and other factors, such as ethnicity and social and lifestyle factors, may also play a role. Sunshine hours, heat, meal distribution and other cultural characteristics affect the cardiac rhythm during the day and may modify the pattern of MI onset as well [29]. Both biological and environmental factors likely contribute to the onset of symptoms and outcomes in acute myocardial infarction. The presence of certain intrinsic (genetic, ethnic, pharmacological, comorbidities, lifestyle, chronotype, culture and/or social) [29,30,31,32] and extrinsic (climatology, built environment) factors may modify this circadian rhythm characteristic of infarction onset in the general population and induce the clinical presentation of AMI in different time zones [17,18,33,34,35].

In the current study, ACS seems to occur more often in the morning hours on Mondays, Tuesdays, Wednesdays and Thursdays than on Fridays to Sundays and in the winter. This result is in agreement with several previous studies that investigated the circadian distribution of acute coronary syndromes [3,6,20,36,37]. This increase in the mornings is reasonable and could be explained by the morning increase in sympathetic activity and plasma cortisol and renin levels, which peak at the time of awakening [1]. Subsequently, the systemic vascular tone, blood pressure and heart rate increased. These changes seem to be related to the probability of plaque rupture, significantly increasing from 06:00–11:59. Plaque ruptures occur most frequently in the morning [18,35].

Moreover, our results showed that the onset of pain in the afternoon or night decreases the influence of the severity of the AMI occurrence, although this was not significant, which was also confirmed by the results of Albackr et al. (2019) [17].

This result could also be connected with the location of AMI. It is well known that anterior infarction is related to the severity of left coronary artery involvement, and it was in the morning when we observed more cases of anterior AMI. Furthermore, the onset of pain at night was associated with the subendocardial location of the infarction as opposed to an anterior location [38]. Our results are consistent with those of Moruzzi et al. (2004), who showed that regarding the incidence of myocardial infarction, the time of onset from 6 am to noon (12 am) was associated with an increased risk of anterior infarction, suggesting the protective role of sleep [39].

Furthermore, these outcomes are also linked with our results that showed that being a smoker increases the likelihood of an inferior AMI. This agrees with the study by Touley et al. (2019), who demonstrated that smoking was independently associated with inferior myocardial infarction [40]. Other authors have also noted that tobacco affects the right coronary system more than the left coronary system [41], thus describing the smoker’s paradox that refers to the “protective” effect of smoking in patients suffering an AMI [42].

Nevertheless, a longer length of hospital stay was associated with patients who presented with the onset of pain in the afternoon. Albackr et al. (2019) noticed that there was an association between the time of symptom onset and the duration of pre-hospital delay, as patients with symptom onset between 18:00 and 23:59 tended to wait longer than patients with symptom onset at other times of the day, especially during working hours [17], and therefore may experience more complications due to the delay in treatment. 

Our sample also consisted of older individuals, which is similar to the study conducted by Tomer et al. (2021), who showed that the patients in the long hospital stay group were the oldest (over 65 years old) with more comorbidities [43].

This study also showed differences in cardiovascular risk factors for the severity of acute myocardial infarction. However, patients divided according to their modifiable and non-modifiable risk factors did not display different behaviours regarding the onset of pain. These results are similar to those obtained in the ARIAM study [8], which showed that regarding age, sex, previous ischaemic heart disease, previous cerebrovascular accident and a family history of ischaemic heart disease (risk markers) and according to arterial hypertension and dyslipidaemia (modifiable factors), the cases conserved the sinusoidal pattern with a single peak in the morning. However, this does not coincide with the pattern described for diabetic patients or smokers, where the authors demonstrated a bimodal pattern of morning and afternoon peaks. This could be influenced by patients’ different chronotypes, as suggested in the study by Selvi et al. (2011), who showed that the peak in morning patients occurred at a different time from that in evening patients [32], and also Škrlec et al. (2018), who found a significant difference in female patients with an intermediate chronotype and diabetes mellitus type 2 [31]. On the other hand, Rouzbahani et al. (2021) did not find any differences in regard to diabetic patients, and in agreement with our results, they found a higher frequency of AMI in the winter [20].

Our study has several limitations. We observed a trend regarding some factors associated with ischaemic heart disease variables. The risk of high severity AMI is greater for participants whose pain starts at the weekend and is lower in seasons other than winter, and regarding the length of hospital stay, the weekdays vs. weekends variable and the season variable did not show a significant association, although the % change on weekends was lower than on workdays and the % change in spring was less than in winter. However, these results did not reach statistical significance due to the small sample size. The sample should be expanded by working with other hospitals in order to be able to compare climatic differences with other regions. It is also important to be able to expand biochemical parameters, such as troponin levels, and circadian chronotypes in order to study their relationships with the circadian rhythm of AMI onset.

## 5. Conclusions

Differences in the circadian rhythm of AMI in different countries were reported. The results of this study indicate that the Spanish people living in a Mediterranean country show a circadian pattern of AMI. ACS seems to occur more often in the morning hours, on Mondays, Tuesdays, Wednesdays and Thursdays, and in the winter. This pattern is observed in all patients and is, therefore, not influenced by patients’ cardiovascular risk factors.

This study also highlights the prognostic relevance of infarction location in relation to circadian rhythm. The onset of pain at night was associated with a subendocardial location of the infarct versus an anterior location.

Morning is associated with an increased risk of anterior infarction, which is related to the severity of the disease. In this sense, being a smoker increases the likelihood of inferior AMI. Therefore, the onset of pain in the afternoon or at night decreases the severity of AMI and anterior infarction is less frequent.

The knowledge of circadian rhythms and the pathophysiology of CVD may lead to new therapeutic approaches. The identification of the cause and timing of the event may allow a more individualised approach to improving prognosis, as well as stricter follow-ups during certain months of the year or the more individualised treatment of patients with coronary artery disease.

## Figures and Tables

**Figure 1 ijerph-19-15628-f001:**
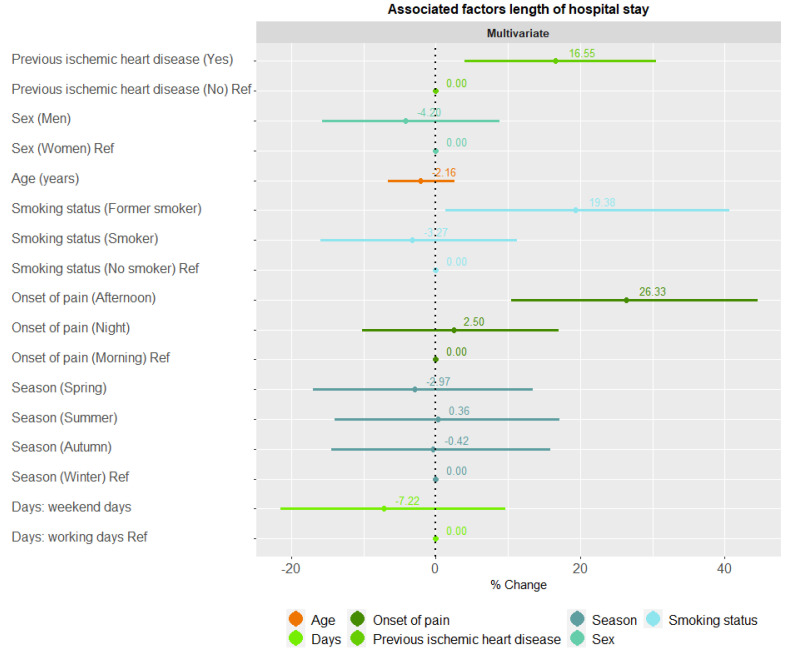
Associated factors with length of stay.

**Table 1 ijerph-19-15628-t001:** Distribution of participants by sociodemographic characteristics, lifestyle and history of disease (N = 244).

Variable		N	%
Sex	Women	71	29.1
	Men	173	70.9
Smoking status	Non-smoker	114	46.7
	Former smoker	43	17.6
	Smoker	87	35.7
Family history of ischaemic heart disease (angina or AMI)	No	222	91.0
	Yes	22	9.0
Previous ischaemic heart disease	No	145	59.4
	Yes	99	40.6
Hypertension	No	122	50.0
	Yes	122	50.0
Dyslipidaemia	No	134	54.9
	Yes	110	45.1
Mellitus diabetes	No	176	72.1
	Yes	68	27.9
Weekday	Sunday	26	10.7
	Monday	42	17.2
	Tuesday	46	18.9
	Wednesday	44	18.0
	Thursday	34	13.9
	Friday	26	10.7
	Saturday	26	10.7
Season	Winter	64	26.2
	Spring	58	23.8
	Summer	59	24.2
	Autumn	63	25.8
Onset of pain	Morning	93	38.1
	Afternoon	74	30.3
	Night	77	31.6
Type of ischaemic disease	Acute myocardial infarction	162	66.4
	Unstable angina	38	15.6
	Other ischaemic heart diseases	44	18.0
Location of acute myocardial infarction	Anterior myocardial infarction	62	38.3
	Inferior myocardial infarction	65	40.1
	IAMNES Subendocardial	22	13.6
	IAM No specified	13	8.0
Severity of ischaemic heart disease	Class I	76	31.1
	Class II	122	50.0
	Class III or Class IV	46	18.9
Age (mean sd)		65.13	13.4
Length of hospital stay (days) (mean sd)		5.09	2.5

**Table 2 ijerph-19-15628-t002:** Associated factors with severity of ischaemic heart disease.

Variable	OR	95%CI *	
Sex (man) (ref. woman)	1.01	0.56	1.83
Age	**1.03**	**1.01**	**1.06**
Smoking status (former smoker) (ref. no smoker)	1.10	0.53	2.29
Smoking status (smoker)	1.25	0.67	2.34
Previous ischaemic heart disease (yes) (ref. no)	**3.26**	**1.90**	**5.58**
Dyslipidaemia (yes) (ref. no)	**1.69**	**1.03**	**2.78**
Onset of pain (afternoon) (ref. morning)	0.88	0.47	1.62
Onset of pain (night)	0.93	0.51	1.68
Season (spring) (ref. winter)	0.76	0.38	1.55
Season (summer)	0.75	0.37	1.55
Season (autumn)	0.90	0.45	1.80
Working days vs. weekend days (ref. workday)	1.54	0.73	3.23

* Variables significant at the 95% CI are highlighted in bold.

**Table 3 ijerph-19-15628-t003:** Associated factors with the location of AMI.

	Anterior AMI	Inferior AMI				Subendocardial AMI			
	%/Mean	%/Mean	OR	CI95% *		%/Mean	OR	CI95% *	
Sex (Man) (ref. woman)	75.8	72.3	0.88	0.36	2.17	59.1	0.36	0.10	1.32
Age	64.2	65.4	**1.03**	**1.00**	**1.06**	64.2	0.99	0.95	1.04
Smoking status (former smoker) (ref. no smoker)	16.1	13.8	0.89	0.27	3.01	22.7	1.41	0.28	7.21
Smoking status (smoker)	32.3	49.2	**2.82**	**1.09**	**7.27**	22.7	0.43	0.10	1.84
Onset of pain (afternoon) (ref. morning)	30.6	33.8	1.38	0.57	3.38	22.7	1.78	0.39	8.03
Onset of pain (night)	21.0	29.2	1.68	0.64	4.41	59.1	**6.87**	**1.67**	**28.15**
Season (spring) (ref. winter)	25.80	29.20	0.79	0.26	2.34	18.2	0.41	0.09	1.90
Season (summer)	25.80	18.50	0.60	0.19	1.91	13.6	0.35	0.07	1.80
Season (autumn)	30.60	26.20	0.63	0.21	1.85	22.7	0.35	0.08	1.47
Working days vs. weekend days (ref. workday)	8.10	18.50	3.27	0.94	11.34	18.2	2.98	0.55	16.11

The reference group for the regression analysis was anterior AMI. AMI: Acute myocardial infarction. * Variables significant at the 95% CI are highlighted in bold.

## Data Availability

Not applicable.

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
