# Peer review of "Circadian Variations and Associated Factors in Patients with Ischaemic Heart Disease"

_ijerph, 2022, doi:10.3390/ijerph192315628_

Round 1

Reviewer 1 Report

The article is overall well-written and easy to understand, however, it would benefit from some light editing for grammatical clarity.

Author Response

Response to Reviewer 1 Comments

Point 1: The article is overall well-written and easy to understand, however, it would benefit from some light editing for grammatical clarity

Response 1: We thank the reviewer’s recommendation. Although the article was reviewed by a first language editing service before submission to the journal, we have sent it to MDPI´s language editing service, as suggested by the reviewer.

Reviewer 2 Report

The manuscript 'Circadian Variations and Associated Factors in Patients with Ischemic Heart Disease' shows the relationship between the occurrence of ischemic heart disease with a circadian rhythm, that is, time of day, day of the week, and season. The manuscript provides insight into a new perspective on the relationship between risk factors and circadian rhythm on the onset of ischemic heart disease. The paper is well-written, and the research results support the conclusions. However, I would like to suggest the following revisions.

The discussion should also include the research results on the relationship between patients' chronotypes and the occurrence of ischemic heart diseases (such as AMI). Valuable articles for comparison are DOI: 10.3109/07420528.2011.559605 and DOI: 10.20471/ACC.2018.57.03.11.

1. Line 76 - write the full name of the abbreviation SIP.

2. Lines 93 - 94 - the authors wrote that the working days are Monday, Tuesday, Wednesday, Thursday, and Friday, and the weekend is Saturday and Sunday. However, in the results (lines 165 - 166), it is written that working days Monday, Tuesday, Wednesday, and Thursday are the days with the highest incidence of ischemic events compared to the days from Friday to Sunday. Therefore, in the results shown in Tables 2 and 3, are working days from Monday to Friday or days from Monday to Thursday taken into the calculation?

3. Line 96 and line 245 - inconsistency in writing comorbidity. Decide on one and stick to it.

4. Table 1 - Family history - what exactly does it refer to, angina, AMI, or all together? It is necessary to specify.

5. Table 1 - Length of hospital stay (day) is shown as mean and SD or as median and IQR? Specify.

6. Line 225 - correct the IMA into AMI.

7. Lines 235 - 236 - it seems the sentence is not completed.

8. Line 247 - if the abbreviation AMI was introduced at the beginning of the manuscript, it should be used throughout it.

9. Line 250 - write the full name of the abbreviation ACVA.

Author Response

Response to Reviewer 2 Comments

Point 1: The discussion should also include the research results on the relationship between patients' chronotypes and the occurrence of ischemic heart diseases (such as AMI). Valuable articles for comparison are DOI: 10.3109/07420528.2011.559605 and DOI: 10.20471/ACC.2018.57.03.11.

Response 1: We thank the reviewer’s recommendation. We have written a paragraph in the discussion section (lines 893-898) and have also included the articles recommended by the reviewer (reference 28 and 41).

Point 2: Line 76 - write the full name of the abbreviation SIP.

Response 2: We thank the reviewer’s indication; it was a mistake. We have removed it.

Point 3. Lines 93 - 94 - the authors wrote that the working days are Monday, Tuesday, Wednesday, Thursday, and Friday, and the weekend is Saturday and Sunday. However, in the results (lines 165 - 166), it is written that working days Monday, Tuesday, Wednesday, and Thursday are the days with the highest incidence of ischemic events compared to the days from Friday to Sunday. Therefore, in the results shown in Tables 2 and 3, are working days from Monday to Friday or days from Monday to Thursday taken into the calculation?

Response 3. We thank the reviewer's comments. Working days refer to days from Monday to Friday and non-working days are Saturday and Sunday.  In Table 1 we refer to the incidences of each day of the week and we comment in the text which days had the highest incidence, while in the association analyses (Table 2, 3 and Figure 1) we decided to group the days into working and non-working days to increase the statistical power, due to the low n in the categories.

Point 4. Line 96 and line 245 - inconsistency in writing comorbidity. Decide on one and stick to it.

Response 4. We thank the reviewer’s suggestion. We have written comorbidity throughout the text.

Point 5. Table 1 - Family history - what exactly does it refer to, angina, AMI, or all together? It is necessary to specify.

Response 5. Thanks the reviewer’s suggestion. This refers to a family history of ischaemic heart disease with angina or AMI . We have added it in the table 1.

Point 6. Table 1 - Length of hospital stay (day) is shown as mean and SD or as median and IQR? Specify.

Response 6. Thanks to the Reviewer for reviewing this point. It is shown as a mean and SD. We have added it in the table 1

Point 7. Line 225 - correct the IMA into AMI.

Response 7. Thanks to the Reviewer for reviewing this point. We have written AMI throughout the text.

Point 8. Lines 235 - 236 - it seems the sentence is not completed.

Response 8. Thanks to the Reviewer for reviewing this point. We have rewritten this sentence.

Point 9. Line 247 - if the abbreviation AMI was introduced at the beginning of the manuscript, it should be used throughout it.

Response 9. Thanks to the Reviewer for reviewing this point. We have written AMI throughout the text.

Point 10. Line 250 - write the full name of the abbreviation ACVA.

Response 10. Thanks to the Reviewer for reviewing this point. We have written the full name of ACVA (previous cerebrovascular accident).

Finally, although the article was reviewed by a first language editing service before submission to the journal, we have sent it to MDPI´s language editing service, in order to improve it.

Reviewer 3 Report

Dear authors, I enjoyed reading your article. It is well structured and clearly written.

There are a number of issues:

- use of non-fresh sources of literature (more than 50% older than 5 years)

- in table 1, when distributed by days of the week, the percentage of patients is different, but the number is the same - 26, this is probably a typo

- picture 1 - very small inscriptions, difficult to read

- absence in the section "materials and methods" of  criteria of ishemic heart disease (most important!)

Author Response

Response to Reviewer 3 Comments

Point 1. Use of non-fresh sources of literature (more than 50% older than 5 years)

Response 1. We thank the reviewer’s recommendation. We have modified and refreshed the bibliography and more than half of it is from the last 5 years.

Point 2. In table 1, when distributed by days of the week, the percentage of patients is different, but the number is the same - 26, this is probably a typo

 Response 2. The Reviewer is totally right. We have now corrected this mistake in the table 1.

 Point 3. Picture 1 - very small inscriptions, difficult to read

 Response 3. Many thanks to the Reviewer for suggesting a letter's size more suitable for read the picture 1. We have changed it.  

 Point 4. Absence in the section "materials and methods" of  criteria of ishemic heart disease (most important!)

 Response 4. Thanks to the Reviewer for reviewing this point. We have added these criteria in the in the section "materials and methods" (lines 197-363).

The criteria of ischemic heart disease  were based on those established by European Guidelines for the management of acute myocardial infarction in patients presenting with ST-segment elevation 2012 [1]. Detection of rise and/or fall of cardiac biomarker values (preferably troponin) with at least one value above the 99th percentile of the upper reference limit and with at least one of the following:

  • Symptoms of ischaemia (history of chest pain lasting for 20 min or more, not responding to nitroglycerine; less-typical symptoms, such as nausea/vomiting, shortness of breath, fatigue, palpitations or syncope.
  • New or presumably new significant ST-T changes or new left bundle branch block.
  • Development of pathological Q waves in the ECG;
  • Imaging evidence of new loss of viable myocardium, or new regional wall motion abnormality;
  • Identification of an intracoronary thrombus by angiography or autopsy.

[1]        Steg, P. G.; James, S. K.; Atar, D.; Badano, L. P.; Lundqvist, C. B.; Borger, M. A.; Di Mario, C.; Dickstein, K.; Ducrocq, G.; Fernandez-Aviles, F.; et al. ESC Guidelines for the Management of Acute Myocardial Infarction in Patients Presenting with ST-Segment Elevation. Eur. Heart J. 2012, 33 (20), 2569–2619. https://doi.org/10.1093/eurheartj/ehs215.

Round 2

Reviewer 2 Report

The authors addressed all the mentioned issues.

Author Response

Response to Reviewer 2 Comments:

Point 1: The authors addressed all the mentioned issues.

Response 1: We thank the reviewer’s input.

Reviewer 3 Report

The authors put correctives, gave a detailed answer to the questions.

However, the answer to question 4 is not fully covered, because criteria for MI are given, which is very important, but there are no criteria for making a diagnosis unstable angina and other ischemic heart diseases (for example from https://doi.org/10.1093/eurheartj/ehz425)

Author Response

Response to Reviewer 3 Comments

Point 1: The authors put correctives, gave a detailed answer to the questions.

However, the answer to question 4 is not fully covered, because criteria for MI are given, which is very important, but there are no criteria for making a diagnosis unstable angina and other ischemic heart diseases (for example from https://doi.org/10.1093/eurheartj/ehz425)

Response 1: We thank the reviewer’s recommendation and for providing the reference[1]. We have inserted this reference in the introduction (line 50) to complete the section on acute coronary syndromes.

We have completed the criteria of ischemic heart disease in “material and methods” section (line 87-100) and have provided the criteria to identify patients with possible unstable angina or other ischaemic heart diseases. The references used are those of the European guidelines of the management of acute myocardial infarction in patients with ST-segment elevation (STEMI) [2] and Non-ST-Elevation Myocardial Infarction (NSTEMI) [3] and stable coronary artery disease [4] because the patients were included in 2015 so the criteria used were prior to the reference suggested by the reviewer.

(1)      Knuuti, J.; Wijns, W.; Saraste, A.; Capodanno, D.; Barbato, E.; Funck-Brentano, C.; Prescott, E.; Storey, R. F.; Deaton, C.; Cuisset, T.; et al. 2019 ESC Guidelines for the Diagnosis and Management of Chronic Coronary  Syndromes. Eur. Heart J. 2020, 41 (3), 407–477. https://doi.org/10.1093/eurheartj/ehz425.

(2)      Steg, P. G.; James, S. K.; Atar, D.; Badano, L. P.; Lundqvist, C. B.; Borger, M. A.; Di Mario, C.; Dickstein, K.; Ducrocq, G.; Fernandez-Aviles, F.; et al. ESC Guidelines for the Management of Acute Myocardial Infarction in Patients Presenting with ST-Segment Elevation. Eur. Heart J. 2012, 33 (20), 2569–2619. https://doi.org/10.1093/eurheartj/ehs215.

(3)      Roffi, M.; Patrono, C.; Collet, J. P.; Mueller, C.; Valgimigli, M.; Andreotti, F.; Bax, J. J.; Borger, M. A.; Brotons, C.; Chew, D. P.; et al. 2015 ESC Guidelines for the Management of Acute Coronary Syndromes in Patients Presenting without Persistent St-Segment Elevation: Task Force for the Management of Acute Coronary Syndromes in Patients Presenting without Persistent ST-Segment Elevation Of . Eur. Heart J. 2016, 37 (3), 267–315. https://doi.org/10.1093/eurheartj/ehv320.

(4)      Montalescot, G.; Sechtem, U.; Achenbach, S.; Andreotti, F.; Arden, C.; Budaj, A.; Bugiardini, R.; Crea, F.; Cuisset, T.; Di Mario, C.; et al. 2013 ESC Guidelines on the Management of Stable Coronary Artery Disease. Eur. Heart J. 2013, 34 (38), 2949–3003. https://doi.org/10.1093/EURHEARTJ/EHT296.
